# Exploring Extracellular Matrix Crosslinking as a Therapeutic Approach to Fibrosis

**DOI:** 10.3390/cells13050438

**Published:** 2024-03-02

**Authors:** Sarah M. Lloyd, Yupeng He

**Affiliations:** AbbVie Inc., 1 North Waukegan Rd., North Chicago, IL 60064, USA; sarah.lloyd@abbvie.com

**Keywords:** extracellular matrix, crosslinking, collagen, fibrosis

## Abstract

The extracellular matrix (ECM) provides structural support for tissues and regulatory signals for resident cells. ECM requires a careful balance between protein accumulation and degradation for homeostasis. Disruption of this balance can lead to pathological processes such as fibrosis in organs across the body. Post-translational crosslinking modifications to ECM proteins such as collagens alter ECM structure and function. Dysregulation of crosslinking enzymes as well as changes in crosslinking composition are prevalent in fibrosis. Because of the crucial roles these ECM crosslinking pathways play in disease, the enzymes that govern crosslinking events are being explored as therapeutic targets for fibrosis. Here, we review in depth the molecular mechanisms underlying ECM crosslinking, how ECM crosslinking contributes to fibrosis, and the therapeutic strategies being explored to target ECM crosslinking in fibrosis to restore normal tissue structure and function.

## 1. Introduction

Fibrosis, or tissue “scarring”, is a complex, often lethal disease process driven by different cell types and affecting multiple vital organs [1,2,3,4,5]. Even though fibrosis contributes to up to 45% of deaths in industrialized areas, there remains a limited understanding of how fibrosis develops over time and how the process could be reversed and resolved [2,6]. Available treatment options for fibrosis are currently minimal and lack disease-modifying efficacy [2]. One key cell type in fibrotic progression is the fibroblast. Upon activation by various local injury signals, fibroblasts transdifferentiate into myofibroblasts and produce excess extracellular matrix (ECM) [2,7,8]. Under normal, homeostatic circumstances, this process is necessary for repairing injured tissues. The deposited ECM can then be remodeled to restore normal tissue structure and function, and the transiently induced myofibroblasts can self-resolve. When the repair process is dysregulated or tissues are repeatedly injured and stimulated, however, the overproduction of ECM by myofibroblasts leads to fibrosis [1,2,3,4,5,7]. This causes reduced levels of oxygen and promotes myofibroblast persistence and disease progression [1,3]. Tissue density, stiffness, and other biophysical features are also altered during increased ECM production which further activates myofibroblasts through mechano-signaling [2,3,5,9]. Overproduction and malformation of ECM is the signature characteristic of fibrosis and creates a positive feedback loop, causing continued progression of fibrosis [3,9,10].

In fibrotic organs, ECM changes not only in abundance, but also in composition [8]. In a healthy state, the ECM is a complex 3D macromolecular network that fills the space between cells in solid tissues. It consists of approximately 300 different core matrisome proteins, which include collagens, proteoglycans, and glycoproteins [11]. The ECM provides not only physical structural support to tissues, but also biochemical and biomechanical signals to regulate numerous cellular functions during homeostasis and pathogenesis [2,11]. In fibrosis, activated myofibroblasts produce excess fibril-forming collagens I and III, as well as fibronectin and elastin [8,10]. The buildup of these proteins can be balanced by ECM-remodeling enzymes, including matrix metalloproteinases (MMPs), adamalysins, or meprins [12]. MMP and adamalysin enzyme activities are regulated by tissue inhibitor metalloproteinases (TIMP) family proteins, which prevent the over-degradation of the ECM [12]. The imbalance between ECM accumulation and remodeling has been highlighted as a recurring feature in fibrosis [10,12].

In addition to regulation by degradation enzymes, ECM proteins are subject to post-translational modifications (PTMs) which alter their structure and function [13,14]. The over-crosslinked, stiffened ECM is a defining feature in fibrotic disorders across various organs, including lung, liver, and skin [13], as well as an important pathological driver of progressive fibrosis. This provides a unique opportunity for therapeutic strategies targeting ECM in fibrosis. Rather than depleting or degrading ECM as a whole, ECM crosslinking can be selectively targeted [13,14,15]. Here, we will review the mechanisms underlying crosslinking modifications, the influence of crosslinking on fibrosis progression, and therapeutic approaches toward ECM crosslinking aimed at normalizing the diseased ECM and treating fibrotic diseases.

## 2. ECM Crosslinking Biochemical Pathways

Crosslinking serves as a powerful regulator of the biophysical properties of ECM. It creates strong connections between ECM molecules, giving them increased stability and resistance to proteolysis [14,16,17]. Crosslinking modifications can be mediated by enzymatic or non-enzymatic reactions. Crosslinks formed via non-enzymatic reactions occur slowly and are associated with aging, making this type of crosslink challenging for therapeutic targeting [14,17]. Enzymatic crosslinking is mediated by multiple groups of enzymes, including lysyl oxidase (LOX) and transglutaminase (TG) proteins [13,14]. While not directly contributing to crosslinking reactions, lysine hydroxylase (LH) enzymes also play a key role in LOX-mediated crosslinking as they modify amino acids that are subsequently acted on by the LOX family [13].

Since collagens are the most abundant fibrous proteins of interstitial tissue ECM and the major constituents of fibrotic ECM, we would like to place particular emphasis on pathways and mechanisms of collagen crosslinking in this section [18]. Collagen goes through a complex sequence of processing steps before reaching its final, mature form in the ECM. After being transcribed, collagen is translated into alpha chains containing N- and C-terminal pro-peptides [19,20]. Proline hydroxylation of these alpha chains then allows for stable formation of a procollagen triple helix [21,22]. Procollagen can be made up of three of the same (homotrimeric), a combination of three different (heterotrimeric), or two of the same and one different (heterotrimeric) alpha chain. When these molecules are initially formed in the endoplasmic reticulum (ER), they still contain N- and C-terminal pro-peptides, which are cleaved after release into the ECM. After the cleavage, these tropocollagen triple helices contain three domains: N-terminal telopeptide, C-terminal telopeptide, and a helical domain in the center of the molecule. Finally, fibrillar collagen molecules can assemble into stable fibrils regulated by intra- and intermolecular crosslinking (Figure 1) [14,20,21,23,24,25]. Next, we will discuss the enzyme families that mediate the modification and crosslinking of ECM.

### 2.1. Transglutaminase (TG) Crosslinking

There are nine different TG genes, of which eight produce catalytically active enzymes [26]. These enzymes regulate a crosslinking reaction between glutamine and lysine residues (Figure 2) [13,26]. Importantly, TG activity is not limited to crosslinking, and TGs have a number of different substrates [13,27]. TGs also serve as scaffolding proteins important for basic cell functions, including cell adhesion and signal transduction [26,28]. Such a diverse family of enzymes requires careful regulation for the maintenance of homeostasis. TG activity is calcium-dependent, which can be advantageous for controlling enzyme activity. Homeostatic cells in the liver maintain inactive TG2 intracellularly, but upon injury, they release TG2 into the ECM, where its activity increases due to higher calcium levels [29,30]. The activity of several TGs can also be activated by proteolytic cleavage or diminished through the binding of purine nucleotides [26].

Transglutaminase 2 (TG2) has been identified as the most broadly expressed TG and is dysregulated in a number of diseases, including fibrosis [14,26,28]. It has been specifically studied in kidney, heart, lung, and liver fibrosis [31]. Beyond its functions in crosslinking, TG2 contributes to fibrosis through its protein binding capabilities. It binds to TGFβ, a key profibrotic molecule, which helps facilitate its conversion from a latent to active form [28,32]. Thus, TG2 could serve as a target for treating fibrosis, but drug delivery and specificity should be carefully considered due to the broad expression and variety of essential functions.

### 2.2. Lysine Hydroxylase (LH)

Three genes encode for three different LH enzymes. While at the protein level these are referred to as LH1, LH2, and LH3, the genes are named *PLOD1*, *PLOD2*, and *PLOD3* (procollagen-lysine, 2-oxoglutarate 5-dioxygenase), respectively [13,24]. The LH family does not catalyze crosslinks, but its activity is crucial for LOX-mediated crosslinking reactions. These enzymes contribute to crosslinking by converting lysine to hydroxylysine on collagens (Figure 3A). LOX enzymes can act on unmodified lysine, but a number of crosslinks are derived from the hydroxylysine generated by LHs [13,21].

LH modifications to collagen are substrate-specific. Hydroxylation of collagen telopeptides is conferred through LH2, which has two isoforms: LH2a and LH2b. LH2b, but not LH2a, is necessary for downstream collagen crosslinking [13,22,24,33]. Helical regions of collagen are hydroxylated by LH1 and LH3 [13,24]. LH1 and LH3 play non-redundant crosslinking functions, because they have specificity for different types of collagens; LH1 plays a dominant role in the helical region hydroxylation of collagens I and III, while LH3 plays a stronger role in modifying collagens IV and V [22]. LH3 additionally has a unique domain conferring galactosyltransferase activity [23]. All three LH enzymes predominantly localize to the endoplasmic reticulum within the cell. LH2 and LH3 have also been identified outside of the cell and have been shown to maintain catalytic activity [23,24,34]. LH enzyme activity is critical for modifying collagens before they are modified by LOX [21].

The differences between LH enzymes are crucial in considering crosslinking in fibrosis. For example, collagens I and III are key markers in lung fibrosis [9,35,36]. Thus, in pulmonary fibrosis, LH1, which preferentially modifies these collagens, may be a more interesting contributor than LH3. However, this only accounts for helical region modifications to collagen; telopeptide region hydroxylation is also critical since it is required for the formation of mature crosslinks [24]. *PLOD2*, the gene encoding LH2, the protein responsible for telopeptide hydroxylation, is upregulated in samples from both idiopathic pulmonary fibrosis and systemic sclerosis patients [37,38,39]. Overall, these LH proteins create an important foundation for LOX-mediated crosslinking.

### 2.3. Lysyl Oxidase (LOX) Crosslinking

The LOX family includes LOX and four LOX-like proteins (LOXL1–4) [40]. Different LOX family proteins play unique roles in cytoskeleton organization and in transcriptional regulation through interactions with histones and transcription factors [40]. Perhaps of more interest to fibrosis, however, LOX proteins mediate crosslinking on collagen and elastin [15]. The crosslinks formed by LOX enzymes are both protein- and region-specific. Collagen crosslinks differ from elastin crosslinks, and crosslinks on collagen vary between helical and telopeptide regions [15,21].

While TGs are calcium-dependent, LOX enzymes are copper-dependent [21,40]. Other regulatory mechanisms for LOX enzymes vary by protein. LOX and LOXL1 are activated by proteolytic cleavage of their N-terminal domains [15,40]. This is performed by bone morphogenetic protein-1 (BMP-1) after they are secreted [40]. Rather than an N-terminal pro-peptide, LOXL2, 3, and 4 have four SRCR repeats [40]. These structural differences provide some context as to the differing functions between these enzymes. Extensive work has provided a comprehensive view of the biochemical reactions underlying LOX crosslinking on both collagen and elastin.

### 2.4. LOX-Mediated Collagen Crosslinking

LOX-mediated collagen crosslinks can be intramolecular or intermolecular and can span across different types of collagens [15]. Unlike TGs, multiple subsequent LOX-mediated reactions are required to achieve crosslinked collagen. The initial chemical reaction triggered by LOX enzymes is the oxidative deamination of a lysine or hydroxylysine, resulting in an aldehyde (Figure 3B) [13]. These aldehydes go through condensation reactions with a lysine or a hydroxylysine [13]. If neither amino acid reacting is modified by an LH enzyme, a lysine aldehyde with a lysine, dehydro-lysino-norleucine (deH-LNL) is produced. When one or more of the reacting molecules is modified by LH, the product includes one or two hydroxyl groups. A lysine aldehyde with hydroxylysine produces dehydro-hydroxylysino-norleucine (deH-HLNL). Similarly, if a hydroxylysine aldehyde reacts with a lysine, the same product is produced, deH-HLNL. Lastly, if both reacting molecules are hydroxylated by LH, a hydroxylysine aldehyde and a hydroxylysine, dehydro-dihydroxylysino-norleucine (deH-DHLNL) is produced (Figure 4) [13,41]. In summary, crosslinks formed by LOX are deH-LNL, deH-HLNL, and deH-DHLNL, where each product varies only in the number of hydroxyl groups, zero, one, or two, respectively.

deH-HLNL and deH-DHLNL are immature, divalent crosslinks and can go through an additional reaction to become mature and trivalent [21,42,43]. This reaction can create links between two to three different collagen molecules [21]. Reactions with various forms of lysine produce pyridinolines or pyrroles. Pyridinoline (Pyr) is formed by the reaction between deH-DHLNL and a hydroxylysine aldehyde. Deoxy-pyridinoline (DPyr) is formed with deH-HLNL and a hydroxylysine aldehyde. While Pyr and DPyr are formed with hydroxylysine aldehydes, pyrroles are formed with lysine aldehydes. deH-HLNL with a lysine aldehyde forms deoxy-pyrrole (DPrl), while deH-DHLNL with a lysine aldehyde forms pyrrole (Prl) (Figure 5) [21,42,44].

According to some studies, collagen crosslinks can also be formed with histidine residues. A crosslink can be formed between deH-HLNL and a histidine residue forming histidino-hydroxylysinonorleucine (HHL) [21,44]. Dehydro-histidino-hydroxymerodesmosine (HHMD) can also be formed through a series of reactions involving lysine aldehyde, hydroxylysine, and histidine [21,44]. In 2019, one group claimed that HHL was just an artifact; however, this claim was disputed by Yamauchi, Taga, and Terajima in a letter to the editor of the same journal [45,46]. This discrepancy may in part be due to a limitation in the methodology used [45]. Thus, it is important to consider methods available for answering specific questions about crosslink modifications. In summary, we have discussed nine unique crosslinking modifications to collagen facilitated by LOX: deH-LNL, deH-HLNL, deH-DHLNL, Pyr, DPyr, Prl, DPrl, HHL, and HHMD. Evaluation of variations in these crosslinking modifications can provide insights into ECM changes in fibrosis.

### 2.5. LOX-Mediated Elastin Crosslinking

LOX, LOXL1, and LOXL2 additionally facilitate crosslinking on elastin protein in the ECM [15,17]. As with collagen, the first step to this crosslinking is an LOX enzyme converting a lysine to a lysine aldehyde in the extracellular space. Subsequently, through a condensation reaction between a lysine aldehyde and a lysine, an immature deH-LNL crosslink is formed [15,17]. Since collagen also has deH-LNL crosslinks, assays measuring deH-LNL could identify these crosslinks from either collagen or elastin. On elastin, lysine aldehydes can also interact with one another to form an allysine aldol (AA). deH-LNL and AA are bifunctional, and as with immature collagen crosslinks, they go through additional reactions to form mature crosslinks. AA interacts with unmodified lysine to generate dehydromerodesmosine (deH-MDES) [17]. deH-MDES can also be formed through the interaction between a lysine aldehyde and deH-LNL. deH-MDES crosslinks are trifunctional. Elastin also has tetravalent mature crosslinks, desmosine (Des) and isodesmosine (IDes), which are formed through the reaction between deH-MDES and a lysine aldehyde [15,17]. These mature, tri-, and tetra-valent crosslinks are unique to elastin. Accumulation of crosslinks on elastin limits proteolytic degradation and stabilizes elastin for long periods of time [17].

### 2.6. LOX in Fibrosis

LOX activity is a critical element in fibrosis and is being investigated as a target for therapeutics [15,47,48]. Like TGs, LOX proteins bind to TGFβ, a key regulator of fibrosis [15]. LOX and LOXL2 are also regulated by hypoxia [15], which mechanistically explains the upregulation of LOX activity in fibrotic tissues where hypoxia is prevalent [1,3,49]. In a recent study by Brereton et al., the relationship between hypoxia-inducible factor (HIF) pathway and LOX activity was explored in depth. The group found that in addition to increased expression of LOXL2, collagen structure, tissue stiffness, and quantity of mature pyridinoline crosslinking were altered upon HIF pathway activation in a lung fibrosis model [39].

Types of crosslinking dynamics observed in fibrosis vary by organ in part due to the tissue-specificity of crosslinking enzymes and modifications [23]. In skeletal tissues, including bone and cartilage, LH2 drives the formation of crosslinks from hydroxylysine aldehydes: HLNL, DHLNL, pyridinoline, and deoxypyridinoline [15,20]. Pyrrole crosslinks are predominantly found in tendons and mineralizing tissue [15,21]. For this reason, pyrroles may have less relevance for research in organ fibrosis. In skin and cornea, crosslinks from lysine aldehydes have been shown to be more prevalent [15,20]. Thus, maintaining appropriate crosslinking composition is necessary for homeostasis and basic organ functionality. When targeting crosslinking to treat fibrosis of one organ, the effect on crosslinking in other organs may also need to be considered if a drug is delivered systemically.

In fibrosis, there is not only an observed increase in LOX expression and overall crosslinking, but also a change in proportions of different types of crosslinks [23]. Crosslinks derived from the hydroxylysine aldehyde pathway (deH-HLNL, deH-DHLNL, Pyr, and DPyr) have been shown to increase in fibrotic conditions [33,49,50,51]. These crosslinks not only alter ECM structure and biomechanical properties, but also change protein susceptibility to degradation by MMPs [14,23]. In summary, the crosslinking problem in fibrosis is created by a synergistic effect of the overaccumulation of crosslinked ECM as well as the increased stability of these crosslinked proteins.

### 2.7. Summary of Crosslinking Biochemistry

Overall, ECM crosslinking is carefully regulated by a number of diverse enzymes. The detailed molecular mechanisms regulated by these enzymes are crucial and thus should be considered for therapeutic targeting. It is important to consider their roles not only in crosslinking, but also in other cellular functions. The TG family, for instance, has eight functional enzymes and a vast number of substrates, so therapeutics for these enzymes would likely influence more than ECM crosslinking. The overall impact of the enzyme being targeted is also crucial and warrants consideration. For example, targeting the LH family would prevent the formation of some crosslinks but not all. This could be beneficial for minimizing toxicity but may limit drug efficacy. LOX targeting, on the other hand, would more broadly impact crosslink formation. Depending on which LOX family member is targeted, this approach could influence both collagen and elastin crosslink formation. Ultimately, applying knowledge of crosslinking biochemical pathways will help to guide drug discovery.

## 3. ECM Crosslinking Analysis Methods

Methods to evaluate crosslinking are limited due to the complexity of modifications and the ECM structures formed. SDS-PAGE and size exclusion chromatography (SEC) can be used to detect the presence of crosslinks, but do not clearly decipher between specific types of crosslinks [52]. To quantify changes in certain crosslinks, some immunoblotting and ELISA assays have been developed. These are, however, limited by antibody specificity for crosslinking modifications. Antibodies can also fail to bind when ECM proteins are tightly aggregated or have altered structures [52,53]. Nevertheless, this method has served as an efficient way to evaluate crosslinking. Two different studies of lung fibrosis in recent years have shown changes in mature collagen crosslinks using an ELISA quantifying total Pyd and DPyd crosslinks [39,50]. An ELISA has also been developed to evaluate elastin Des/IDes crosslinks [54,55].

For more quantitative and specific analyses of crosslinking, liquid chromatography (LC) paired with mass spectrometry (MS) approaches have been developed [56]. Typically, this approach requires proteins in the samples to be broken down to the amino acid level through hydrolysis [56]. High-performance liquid chromatography (HPLC) or ultra-performance LC (UPLC) is then used for separation, and mass spectrometry is used for quantification [52,53]. This type of method can be used to simultaneously detect multiple crosslinking modifications, both immature and mature [49,57,58]. For comparison across samples, normalization is often required. This can be performed using a colorimetric assay on hydrolysate to measure total hydroxyproline [55]. Effective identification of crosslinks with this LC-MS is dependent on the availability of standards [52]. Since the sample is hydrolyzed into individual amino acids, this method is also limited by its inability to identify the sites of crosslinks on proteins [52,53]. Overall, both ELISA and LC-MS strategies for quantifying crosslinking modifications are effective but still have limitations.

For research focused on crosslinking therapeutics, both immunoassay and LC-MS methods have value. Obtaining comparable results from both assays could provide more confidence in results. For example, a LOX inhibitor could be assessed through both an ELISA for PyD and DPyd and LC-MS. This is, however, dependent on the availability of LC-MS instrumentation and those with expertise to run crosslinking analysis. ELISAs provide a more accessible alternative, but are not available for all crosslink modifications. Unlike ELISAs, one LC-MS run can simultaneously quantify both immature and mature collagen crosslinks. Thus, methodology must be chosen based on feasibility as well as desired output.

## 4. Dysregulation of ECM Crosslinking in Fibrotic Disease

The biochemical nature of ECM crosslinking and the enzymes that contribute to it are generally shared across organ systems. Expression patterns and crosslinking compositions, however, can vary [15]. Extensive work has been conducted to evaluate crosslinking enzymes and modifications in fibrosis of various organs. We will briefly touch on the nature of fibrosis and crosslinking in lung, liver, and skin, three representative organs of high interest and with large unmet medical needs for fibrotic diseases.

### 4.1. Lung Fibrosis

Idiopathic pulmonary fibrosis (IPF) is typically diagnosed in patients around the age of sixty-five and is more common in men. Upon diagnosis, the patients’ survival time is three to five years on average [59]. The two FDA-approved therapeutics, nintedanib and pirfenidone, only slow progression and do not significantly change prognosis [59]. The PLOD/LH family is dysregulated in IPF. PLOD genes are upregulated in IPF patient serum samples, and *PLOD2* was shown to be the most upregulated family member [38]. Leveraging spatial transcriptomics data, *PLOD2* gene expression was also shown to increase at active sites of fibrosis in IPF tissue [39]. Interestingly, research by Jones et al. assessed isoform-specific changes in LH2 in IPF tissue and did not demonstrate a significant increase in LH2b, the isoform responsible for facilitating collagen crosslinking [50]. Given isoform specific functions of LH2, this raises an important consideration that gene expression data alone may not be sufficient. The synthesis of information across studies, at both transcript and protein levels is critical for identifying strong therapeutic targets.

Beyond the PLOD/LH family, LOX proteins have also been investigated in IPF. Jones et al. showed differential gene expression of *LOXL2*, *LOXL3*, and *LOXL4*, but not *LOX* and *LOXL1* [50]. Brereton et al. and Ma et al., however, showed an upregulation of all five LOX genes in IPF tissue [39,49]. Yet, another study by Tjin et al. (2017) assessed two different IPF datasets and found that *LOXL1* was upregulated in both, but *LOXL2* was only upregulated in one [60]. This group followed up on the gene expression studies with an imaging approach and demonstrated that at the protein level, LOXL1 and LOXL2 were upregulated in IPF tissue [60]. Strengthening these expression studies, LOX enzyme activity measured by amine oxidase was also increased in IPF tissue [50]. Collectively, these investigations suggest that LOX is associated with fibrosis progression in lung.

Beyond changes in just LOX, recent work shines a spotlight on crosslinking changes in lung fibrosis. Jones et al. showed an increase in both immature (deH-HLNL, deH-DHLNL) and mature (Pyr, DPyr) crosslinks in IPF patient lung tissue. There was a higher ratio of deH-DHLNL to deH-HLNL in IPF tissue [50]. Consistent with this, Ma et al. evaluated crosslinking in the bleomycin mouse model of lung fibrosis. They observed a strong increase in deH-DHLNL. A subtle change in Pyr was seen but was not significant when normalized by total collagen [49]. In summary, both crosslink enzyme expression and crosslinking modifications are dysregulated in lung fibrosis.

### 4.2. Liver Fibrosis

The liver is subject to developing fibrosis, especially in cases of chronic wound repair. This wound repair process can be overactivated, particularly under circumstances involving viral infection and alcohol abuse [61,62]. Fibrosis of the liver is reversible and can be treated in early stages, but if left unchecked can progress to an irreversible state, cirrhosis [62]. Both collagen and elastin crosslinking have been shown to change in liver fibrosis. LOX gene expression increases in liver fibrosis and cirrhosis [63]. MMP family members and their inhibitors, TIMPs, also have dysregulated expression [61]. Along with enzyme expression changes, studies have highlighted changes to crosslinking modifications in liver fibrosis. In patient-derived liver samples, pyridinoline crosslinking was increased in both viral hepatitis and cirrhosis [64]. To model liver fibrosis, carbon tetrachloride (CCl_4_) is used to induce inflammation [61]. In one study using a fibrotic mouse CCl_4_ injection model, pyridinoline collagen crosslinking increased in fibrosis and cirrhosis. This study also showed an increase in elastin desmosine crosslinking in cirrhosis [63]. Lastly, recent work has highlighted non-enzymatic, advanced glycation end-product (AGE)-mediated crosslinking in cirrhosis ECM. Targeting this diseased ECM showed promise for limiting fibrosis progression [65].

### 4.3. Skin Fibrosis

Systemic sclerosis (SSc) is an autoimmune, multi-organ fibrotic disease affecting connective tissues, including the skin. Using autopsy samples from an SSc patient, one group showed an increase in Pyd crosslinks in skin, endocardium, fascia, and bladder [51]. With a bleomycin mouse model of skin fibrosis, however, another group failed to identify changes in Pyd, despite histological validation of the model. Upon removal of bleomycin, the fibrosis in this model is reversible [66]. Thus, it is likely that in this case, the bleomycin mouse model did not fully recapitulate irreversible fibrotic changes that occur in human disease. As an alternative to a mouse model, Huang et al. developed self-assembling stromal tissues (SASs) and human skin equivalents (HSEs) using dermal fibroblasts derived from SSc patient skin or from normal healthy control donors [67]. They showed an increase in LOXL4 expression at the mRNA and protein levels. Using a CTX-I ELISA kit, they also saw an increase in crosslinked C-telopeptide type i collagen in SSc-derived cultures relative to normal healthy controls [67]. This is consistent with the increase in mature crosslinks observed in tissue from SSc patients [51]. Crosslinking has additionally been investigated in lipodermatosclerosis (LDS). As shown with SSc, an increase in crosslinking was observed in fibrotic skin from patients with LDS [68]. These studies collectively provide evidence of dysregulated crosslinking in fibrotic skin conditions.

### 4.4. Insights into Crosslinking Dysregulation in Fibrotic Disease

It is clear that crosslinking pathways are dysregulated in fibrotic diseases across organ systems. With this knowledge, therapeutic strategies can be developed to apply the same pharmaceutical across different indications. For example, a LOX inhibitor developed for IPF may also be promising for treating SSc. It is noteworthy, however, that since the baseline levels of crosslinking vary by organ, the toxicity and efficacy of drugs may also differ across fibrotic diseases. Additionally, the prevalence of crosslinking dysregulation in fibrosis could be considered for biomarker development. Since crosslinking alters the structure and function of ECM, changes to crosslinking could serve as a useful readout for disease progression and drug efficacy.

## 5. ECM Crosslink-Based Therapeutic Strategies, Drug Targets, and Molecules

Due to strong evidence for dysregulated ECM crosslinking in promoting fibrosis and the key roles of ECM crosslinking enzymes, therapeutics are actively being investigated to target TG, LOX, and LH proteins, with the most advanced drug molecules in phase I/II clinical trials (Table 1) [69,70,71,72,73,74,75,76,77,78,79,80,81,82,83,84,85,86,87,88,89,90,91,92]. LOX therapeutics vary based on which LOX they target, and both small-molecule and antibody-based targeting approaches have been employed. One strategy has been to solely target LOXL2. For example, Simtuzumab is an antibody targeting LOXL2, but it has failed to improve fibrosis in clinical settings and has been discontinued [69,71,93]. The strategy of targeting LOXL2, however, continues with several small molecules currently in clinical trials. PAT-1251, for example, is now in a phase II clinical trial for myelofibrosis [76]. Another small molecule, GB2064, is also in phase II for myelofibrosis, and results suggest disease-modifying activity [77]. These small-molecule LOXL2 inhibitors may also have the potential to treat other fibrotic diseases. This type of strategy, targeting only select LOX proteins, could be effective for ameliorating fibrosis progression without fully ablating functional collagen crosslinking. On the other hand, it is noteworthy that PXS-5505, a pan-LOX inhibitor, is also in phase 1/2a trials for myelofibrosis. It was reported to be well tolerated and showed preliminary signs of modification to disease [94]. It is too early to tell which LOX-targeting strategy would eventually work for fibrotic diseases, but a variety of approaches are being explored.

TGs and LHs have been targeted to a lesser extent than LOX proteins. Of the TGs, TG2 has been the primary target of interest. Among the multiple small-molecule inhibitors of TG2, ZED1227 is the most advanced and has entered clinical development. While ZED1227 has been investigated for the treatment of celiac disease, it also entered clinical phase II in 2022 for treatment of non-alcoholic fatty liver disease (NAFLD) with significant fibrosis [82,83]. It is noteworthy that inhibitory antibodies have also been discovered to target TG2 [84,85]. The most advanced anti-TG2 antibody, Zampilimab, is in clinical trials for adult kidney transplant patients with chronic allograft injury, with potential follow up interest in fibrotic diseases (NCT04335578). For targeting LH2, small-molecule inhibitors have been discovered and proposed for use in fibrosis and cancer metastasis [90,91,92]. Overall, significant efforts are being made to develop novel therapeutics targeting crosslinking for fibrosis treatment. Continuing these efforts will be critical for determining if targeting collagen crosslinking can successfully treat fibrosis.

## 6. Discussion

Fibrosis across organ systems has proven to be extremely complex, dynamic, heterogeneous, and difficult to effectively target therapeutically. Finding consistencies across different types of fibrosis could serve as an effective strategy to build therapeutics applicable to multiple conditions [3]. One recurring similarity identified across organs is profibrotic adaptations of the ECM. Targeting and modifying the profibrotic ECM may offer a parallel approach to fibrosis therapy in addition to targeting disease-promoting cells in fibrotic tissues. With ECM targeting, we must account for both the cellular and molecular changes producing the ECM as well as the profibrotic ECM itself [5]. As for the profibrotic ECM, we must then consider both ECM protein expression profiles and the post-translational crosslinking modifications that define its structure and function.

An important consideration with respect to fibrotic ECM is the progressive, heterogeneous nature of fibrotic diseases [2,9]. Because of this, it would be interesting to explore spatial and temporal dynamics of crosslinking in organ fibrosis. To do this, the field will need to overcome the bottleneck of limited sample availability from patients. Additionally, the advancement of spatially resolved methodology to assess collagen crosslinking will be required. Methods for assessing crosslinked, mature ECM are in part limited by the insolubility and complex three-dimensional structures formed [52]. Current LC-MS approaches are largely limited to quantification of crosslinking modification products from hydrolyzed samples where proteins are broken down into individual amino acids. This makes it impossible to determine the original site of crosslink modification [53]. Mapping out the three-dimensional ECM crosslink patterns and structures across organs and diseases is an underexplored field that will require breakthrough technologies. 

Of almost equal importance as the analysis of ECM crosslinking is the modeling of ECM crosslink patterns associated with homeostatic and pathological processes to meet basic biology and drug discovery research needs. Limited sample availability from patients, especially from the early stages of fibrosis, creates the requirement for laboratory-based fibrotic models to explore novel therapeutic targets. Many groups have utilized biochemical, cell culture, and animal models to characterize ECM crosslinking alterations associated with fibrosis and to support drug discovery programs [49,50,51,68]. However, the current biology models have significant limitations and do not support robust understanding and targeting of ECM for fibrotic diseases. For example, cell culture models, a popular backbone platform to many biomedical research fields, are often two-dimensional and lack the ability to generate a mature, three-dimensional ECM similar to that found in natural tissues [7,96,97]. Because of limitations to existing models, widespread efforts are being made to develop novel three-dimensional strategies, incorporating a more physiologically relevant ECM [97,98,99]. With the generation of new model platforms, cost and throughput may be a more limiting factor. Ultimately, improved modeling of ECM dynamics will advance our ability to develop and test fibrotic therapeutics.

Considering why therapeutics thus far have failed may help drive efforts moving forward. For example, Simtuzumab, an antibody targeting LOXL2, was discontinued from clinical trials due to a lack of efficacy. It is possible that small molecule therapeutics with broader activity could be more efficacious. Drug distribution and penetration of different therapeutics could also play a role. If therapeutics continue to fail despite efforts to diversify therapeutic platforms, additional strategies could be considered. For instance, instead of targeting just the crosslinking enzymes, the diseased, accumulated ECM could be targeted. Delivery of ECM-degradation enzymes such as MMPs has been suggested and explored, but the approach can be limited by side effects, so more selective delivery approaches may be necessary [14]. Cellular therapy approaches based on cell types possessing ECM-modifying and restoration capacities, such as mesenchymal stem cells and fibrolytic macrophages, may offer an additional strategy [10,95,100,101,102]. How to target and normalize diseased ECM and restore healthy tissue structure and function may be the biggest challenge of this research field. Ultimately, pairing therapies that can manage diseased ECM with therapies that can subsequently prevent further ECM buildup may provide a promising strategy.

## 7. Conclusions

In summary, ECM crosslinking dysregulation is prevalent in fibrotic diseases. This is driven by TG, LH, and LOX enzymes, making them potential therapeutic targets. For drug development, it is key to consider each enzyme family’s unique roles and diverse substrates. The continued development of methodology for evaluating crosslinking will facilitate more predictive and thorough evaluation of therapeutics targeting these pathways. With a number of therapeutics currently in preclinical and clinical trial stages, targeting crosslinking provides a promising strategy for treating fibrotic diseases. 

## Figures and Tables

**Figure 1 cells-13-00438-f001:**
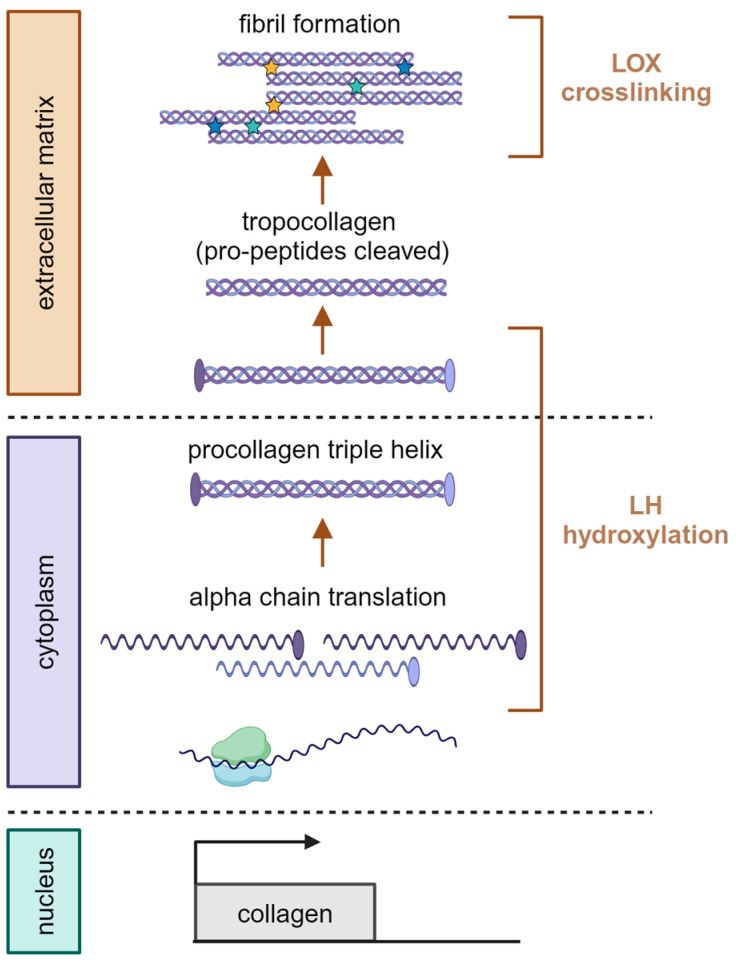
Collagen processing pathway, from transcription to crosslinked fibril formation. After transcription in the nucleus and subsequent translation, three alpha chains assemble into a procollagen triple helix. Upon export into the extracellular space, N- and C-terminal pro-peptides are removed from procollagen, converting it to tropocollagen. For fibrillar collagens, these tropocollagens then form fibrils, which are subject to LOX-mediated crosslinking. LH hydroxylation precedes LOX activity and has been reported to occur both intra- and extracellularly. Ovals indicate N- and C-terminal pro-peptides. Stars are indicative of crosslinks. Created with BioRender.com, accessed on 30 January 2024.

**Figure 2 cells-13-00438-f002:**
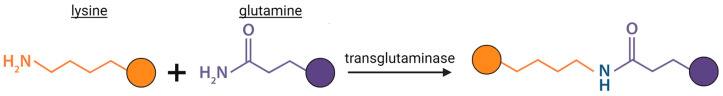
Transglutaminase crosslinking. Transglutaminases (TGs) mediate a crosslinking reaction between lysine and glutamine residues. Created with BioRender.com, accessed on 26 January 2024.

**Figure 3 cells-13-00438-f003:**
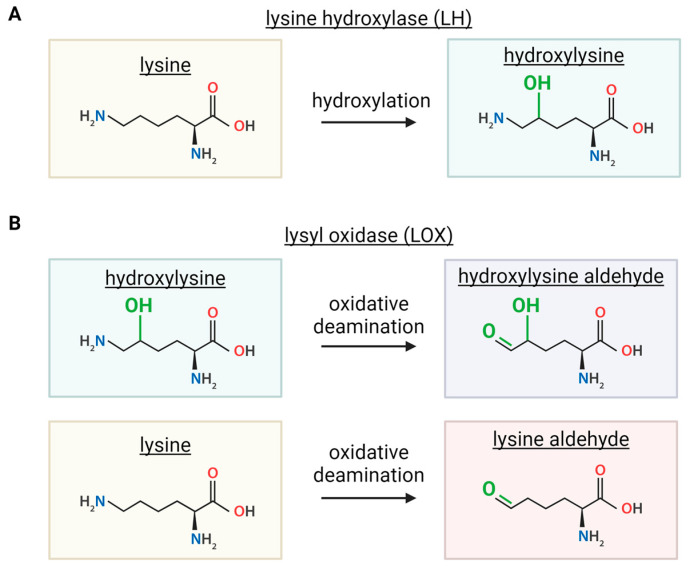
Enzymatic reactions driven by lysine hydroxylase (LH) and lysyl oxidase (LOX) enzymes. (**A**) LHs convert lysine to hydroxylysine through hydroxylation reaction. (**B**) LOXs convert hydroxylysine or lysine into aldehyde form via oxidative deamination. Created with BioRender.com, accessed on 30 January 2024.

**Figure 4 cells-13-00438-f004:**
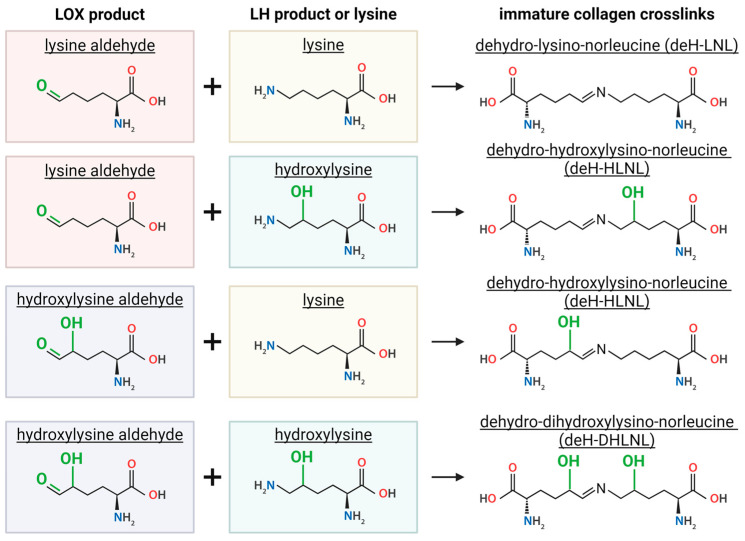
Condensation reactions drive formation of immature collagen crosslinks. A lysine or hydroxylysine aldehyde from a LOX modification reacts with unmodified lysine or a hydroxylysine derived from an LH modification to form three different immature crosslinks. Immature collagen crosslinks vary by number of OH groups (green). Created with BioRender.com, accessed on 30 January 2024.

**Figure 5 cells-13-00438-f005:**
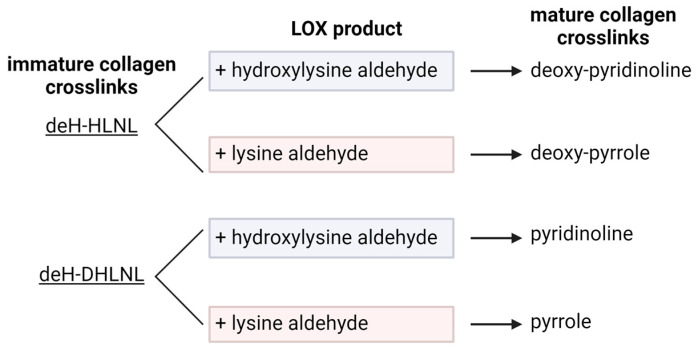
Immature collagen crosslinks go through additional reactions to form mature crosslinks. Pyrroles are formed through a reaction between dehydro-hydroxylysino-norleucine (deH-HLNL) or dehydro-dihydroxylysino-norleucine (deH-DHLNL) with a lysine aldehyde. Pyridinolines are formed through a reaction between deH-HLNL or deH-DHLNL with a hydroxylysine aldehyde. Reactions with deH-HLNL produce the deoxy forms deoxy-pyridinoline and deoxy-pyrrole. Created with BioRender.com, accessed on 26 January 2024.

**Table 1 cells-13-00438-t001:** Therapeutic approaches to targeting ECM crosslinking for fibrotic diseases.

Target	Target	Drug Molecule	Drug Type	Indication	Status	Reference
LOX	LOXL2	Simtuzumab/GS-6624/AB0024	antibody	Cancer, IPF and liver fibrosis, myelofibrosis, PSC, NASH	Phase II—Discontinued	[69,70,71,72]
	Pan-LOX	PXS-5505/SNT-5505	small molecule	Myelofibrosis, liver and pancreatic cancer	Phase II	[73,74,75] NCT05109052, NCT04676529
	LOXL2	PAT-1251	small molecule	IPF and other fibrotic diseases, myelofibrosis	Phase II	[76], NCT04054245, NCT02852551
	LOXL2	GB2064	small molecule	Myelofibrosis	Phase II	[77], NCT04679870
	LOXL2	PXS-5338	small molecule	NASH, IPF, liver and kidney fibrosis	Phase I	[78]
	LOXL2	PXS-5382/SNT-5382	small molecule	Anti-fibrotic IPF/CKD/NASH	Phase I	[75]
	Pan-LOX	PXS-6302	small molecule	Anti-scarring; burns, established scars	Phase I	[79]
	LOXL2/LOXL3	PXS-5153A	small molecule	Liver fibrosis, myocardial infarction	Preclinical	[80]
	LOX	PXS-LOX_1 and PXS-LOX_2	small molecule	Primary myelofibrosis (PMF)	Preclinical	[81]
TG	TG2	ZED1227/TAK-227	small molecule	NAFLD with significant fibrosis	Phase II	[82,83], NCT05305599
	TG2	Zampilimab	antibody	Adult kidney transplant recipients with chronic allograft injury	Phase I/II	NCT04705350, NCT04335578
	TG2	AB1, DC1, and BB7	antibody	Fibrosis and auto-immune disease	Preclinical	[84,95]
	TG2	1–155	small molecule	IPF, cardiac fibrosis	Preclinical	[86,87]
	TG2	R281	small molecule	IPF	Preclinical	[86]
	TG2	GK921	small molecule	Pulmonary fibrosis	Preclinical	[88]
	TG2	Compound 3h	small molecule	Hypertensive nephrosclerosis	Preclinical	[89]
LH	LH2	1,3-Diketone analogs	small molecule	Cancer metastasis	Preclinical	[90,91,92]

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
