# Peer review of "Exploring Extracellular Matrix Crosslinking as a Therapeutic Approach to Fibrosis"

_cells, 2024, doi:10.3390/cells13050438_

Round 1
Reviewer 1 Report
Comments and Suggestions for Authors
In this review, Llyod and He provide a detailed summary on extracellular matrix (ECM) crosslinking, focusing particularly on collagen as it is the main culprit in fibrosis. In addition to describing the enzymes involved in crosslinking the ECM and the resulting molecular crosslinks, they provide examples of fibrotic diseases where these enzymes and crosslinks have been dysregulated. Lastly, they provide an up-to-date table and summary of the small molecule inhibitors and antibodies currently in clinical trials to target ECM crosslinking as a therapeutic strategy. Overall, the review is very well written, fairy comprehensive and timely. Below are my minor comments.
1. In figure 1, it would be helpful to include an annotation for the symbols, i.e. ovals indicate N- and C-domains of collagen, while stars indicate crosslinks.
2. When discussing methods for ECM crosslinking analysis, the authors should include the methods described in the following references. (https://pubmed.ncbi.nlm.nih.gov/29310773/,https://pubmed.ncbi.nlm.nih.gov/29310774/)
3. Minor typos:
a. Line 43, “includes” should be “include”
b. Line 56, “orans” should be “organs”
c. Line 178, “an LH” should be “a LH”
d. Line 321, need to add “was” between “oxidase” and “also”
e. Line 375, replace “the clinic” with “clinical trials” because “in clinic” implies that they are being prescribed.
Reviewer 2 Report
Comments and Suggestions for Authors
This review explores the topic of extracellular matrix crosslinking and the underlying mechanisms. The extracellular matrix is integral to fibrosis, the topic in which this review focuses the therapeutic potential. The review provides a nice intro and gives good insights into the field and the challenges. A few things could potentially benefit this manuscript and are listed below:
11.) Section 2 gets a bit tedious and reads more like a textbook in portions.
22.) Many different enzymes are considered, and mechanisms discussed in section 2, but little new information or insight is present on how the authors suggest the mechanisms could be taken advantage of in a therapeutic sense. It would benefit with some thoughts on potential avenues as each section ends with a version of “these studies show evidence of dysregulation” but provide little author insights. This continues in section 3 and 4.
33.) Why were the 3 examples of fibrosis in section 4 chosen? Would other versions of fibrosis be appropriate to examine?
44.) The discussion seems out of place and could occur within the particular sections (2-4). However, the manuscript could absolutely benefit from a short conclusion at the end.
